# Self-supervised Vision Transformers for Prostate Cancer Classification in Biparametric MRI

Shebna Rose D. Fabilloren[1][0009−0005−4531−3283], Jose Conrado T. Paulino[2][0009−0009−2558−2135], Johanna Patricia A. Cañal[2][0000−0001−5407−1478], and Prospero C. Naval, Jr.[1][0000−0001−7140−1707]

[1] University of the Philippines Diliman, Quezon City, Philippines
[2] University of the Philippines Manila, Manila City, Philippines

**Abstract.** Multiparametric and biparametric magnetic resonance imaging (mpMRI/bpMRI) play an essential role in the detection, pre-biopsy planning, and staging of clinically significant prostate cancer (csPCA). One of the most commonly used structured reporting schemes in the evaluation of prostate MRI's for suspected prostate cancer is the Prostate Imaging–Reporting and Data System (PI-RADS) v.2.1, developed by multiple international representative groups. Existing machine learning models for classifying csPCa using PI-RADS are not reproducible due to the availability of data sets. Meanwhile, public datasets lack PI-RADS labels, a standard in prostate MRI. This hinders progress in the research community. FastMRI Prostate is a recently released, publicly available slice-level MRI dataset with PI-RADS labels. However, research using it is limited due to its recent release, and no studies have yet applied DINOv2 for csPCa classification on bpMRI. Several medical imaging studies have shown DINOv2 to be an effective feature extractor. This study aims to address these gaps by assessing the advantages and limitations of the DINOv2 family of foundation models on the FastMRI Prostate dataset for binary csPCa classification. Our findings reveal that DINOv2 models outperformed other ImageNet pretrained CNN-based models. ViT-g variant obtained an AUROC = 0.889 for the T2W model and 0.862 for the DWI model. This suggests DINOv2 features representations are adaptable to this downstream task. There was minimal performance difference between ViT-g and ViT-L, but a two-fold difference in training time and VRAM needed, making it a good alternative when computational resources are limited. ViT-S (21M parameters) achieved comparable performance to ResNet-152 (60M parameters). Overall, this suggests that DINOv2 models offer a good trade-off between performance and computational cost, making them a viable option even in resource-constrained environments.

**Keywords:** prostate cancer · vision transformer · biparametric mri

## 1 Introduction

Prostate cancer is a common health issue among the male population often present in men over 50 years old [4]. According to the Global cancer statistics

(GLOBOCAN) in 2022, 1.5 million new cases of prostate cancer were registered worldwide [2]. This represents 7.3% of all cancers in men which made it rank second most common cancer. In terms of mortality, it is the fifth leading cause of cancer death in men worldwide. Male patients often seek medical consultation for lower urinary tract symptoms (LUTS), such as increased frequency, urgency, weak or intermittent stream, or difficulty emptying the bladder. Some present with more concerning signs like hematuria, anuria, or dysuria. Others may be asymptomatic but undergo prostate cancer screening based on age-specific clinical guidelines.

The first screening method for prostate cancer screening, after proper history taking and physical examination, is a digital rectal exam (DRE), where a medical professional inserts a finger into the rectum and palpates the prostate gland to provide a rough size measurement and to feel for irregularities in the prostate. However, many experts do not recommend this due to limited evidence of its benefits [22, 15, 11]. The second method is a blood test to measure prostate-specific antigen (PSA) levels. Elevated serum PSA levels may indicate prostate cancer, but may also show elevated results from non-cancerous conditions such as benign prostatic hyperplasia (BPH), which causes prostate enlargement, infections, or due to expected senescent changes, among other causes [13]. Patients with abnormal DRE or PSA results may then be referred for biopsy to confirm diagnosis, as a formal diagnosis of prostate cancer can only be done through histopathologic assessment after biopsy.

Transrectal ultrasound (TRUS)-guided prostate biopsy is a widely used diagnostic procedure for the detection of prostate cancer, typically performed in patients with elevated prostate-specific antigen (PSA) levels or abnormal digital rectal examination findings. Under real-time ultrasound guidance, tissue samples are systematically obtained—usually 10 to 12 cores—from different regions of the prostate for histopathologic evaluation. A pathologist then grades the samples using the Gleason scale [14]. As an invasive procedure, biopsy carries risks such as bleeding, pain, and infection. TRUS-guided biopsy also has a high false-negative rate due to blind sampling, as it often misses areas like the anterior gland, apex, and transition zone.

Pre-biopsy magnetic resonance imaging play an increasingly vital role in the early diagnosis of prostate cancer, and have been recommended prior to biopsies to avoid possible complications as well as false negative sampling. Multiparametric MRI (mpMRI) and biparametric MRI (bpMRI) are both used in prostate cancer imaging, each with distinct advantages and limitations. mpMRI combines T2-weighted (T2W), diffusion-weighted (DWI), and dynamic contrast-enhanced (DCE) imaging, offering high diagnostic accuracy, especially for clinically significant cancer. DCE improves lesion characterization in equivocal or small cases and is supported by clinical guidelines. However, mpMRI requires contrast administration, increasing scan time, cost, and risks in patients with renal insufficiency and may cause major adverse effects, including respiratory or cardiovascular issues and nephrogenic systemic fibrosis. In contrast, bpMRI omits DCE and relies only on T2W and DWI sequences. This approach signifi-

cantly shortens the examination time, reduces costs, and eliminates the need for contrast, making it more accessible and safer for certain patient populations [10]. By using bpMRI, these issues are resolved while still producing similar results with mpMRI [5, 3, 23, 16]. A limitation of bpMRI is its usage with the PI-RADS scoring system, specifically in the evaluation of the peripheral zone lesions with a DWI/ADC score of 3, which elevates to a score of 4 if significant contrast enhancement is detected. Prostate lesions detected on MRI can be graded using the Prostate Imaging–Reporting and Data System (PI-RADS) v2.1. This reporting system aims to standardize prostate MRI acquisition, interpretation, and reporting. The v2.1 scoring system ranges from 1 (very low likelihood of clinically significant prostate cancer, or csPCa) to 5 (very high likelihood). Scoring is based solely on MRI findings and excludes clinical history, digital rectal examination (DRE), and PSA levels.

Deep learning models require each input to have an expected outcome value, also known as a label, to learn the patterns behind the data. In terms of prostate MRI image PI-RADS scoring, the image dataset must contain enough samples for every PI-RADS score (i.e 1 to 5) in order to learn the representation of each score. As highlighted in [1], it is important to have a benchmark dataset to ensure that research published can be beneficial to the public health community. Reproducible research by having a benchmark dataset is essential for advancing machine learning research in medical imaging.

Manual annotation of prostate MRI datasets is time and resource intensive, especially given the limited number of radiologists trained in this specialized type of imaging. The recently released FastMRI Prostate [21] dataset, which is publicly available, includes bpMRI scans with PI-RADS labels and slice-level annotations that reflect how radiologists assess the likelihood of clinically significant prostate cancer (csPCa). While the dataset has not yet been applied to prostate cancer classification tasks, its authors demonstrated feasibility of diagnosing csPCa by training a ConvNeXt [12] binary classification model on the provided slice-level labels. Huang et al. [9] showed that DINOv2 models performed best across three medical imaging benchmarks (i.e Chest X-ray, iChallenge-AMD, HAM10000). This good performance indicates that features learned in the DINOv2 self-supervised training from a huge amount of data can be used for medical imaging. While DINOv2 has shown promise, its transferability to highly specialized modalities like MRI remains an open question. This makes it a viable option for testing vision foundation models on the FastMRI Prostate dataset. This research aims to address the lack of prior work done on binary csPCa classification using slice-level PI-RADS score labels on the prostate bpMRI. The primary contributions of our research are as follows:

- Application and limitations of DINOv2 on prostate MRI: We investigated all DINOv2 model sizes for the binary csPCa classification based on PI-RADS score labels and provide novel insights into its transferability, performance scaling, and domain-specific limitations in prostate MRI.

– Utilization of FastMRI Prostate dataset: We trained a classifier head on top of DINOv2 and CNN-based model backbones that can serve as future baseline performance on this novel dataset.

## 2    Methodology

### 2.1    Data

The FastMRI Prostate [21] dataset was used for linear probing various pretrained models such as DenseNet121 [8], ResNet152 [7], and VGG19 [20] pretrained on ImageNet1k [6], and DINOv2 [17] pretrained on a large, diverse, curated dataset of 142 million images via self-supervised learning. It contains T2W and DWI MRI sequences with PI-RADS labels indicating the existence and score of prostate cancer for each slice. In total, there are 312 subjects which are divided into training, validation, and test groups containing 218, 48, 46 subjects respectively. Table 1 shows the corresponding number of slices for each group including their distribution across PI-RADS categories (P=1 to P=5). For DWI, we only used the ADC map and b1500 DWI sequences.

**Table 1.** Distribution of MRI slices across data splits and PIRADS categories (abbreviated as P=1 to P=5) for T2W and DWI sequences.

| MRI Seq. | Data Split | Total Slices | P=1 | P=2 | P=3 | P=4 | P=5 |
|---|---|---|---|---|---|---|---|
| T2W | Train | 6,647 | 6,106 | 200 | 133 | 88 | 120 |
|  | Validation | 1,462 | 1,345 | 23 | 67 | 10 | 17 |
|  | Test | 1,399 | 1,290 | 41 | 31 | 10 | 27 |
| DWI | Train | 13,274 | 12,186 | 244 | 352 | 242 | 250 |
|  | Validation | 2,916 | 2,672 | 38 | 142 | 18 | 46 |
|  | Test | 2,790 | 2,578 | 46 | 78 | 56 | 32 |

### 2.2    Preprocessing

We trained two models for each MRI sequence. This is patterned after how clinicians use PI-RADS grading when assessing MRI sequences. Figure 2 shows a high-level flowchart of our training approach. In the T2W model, U-Net segmentation was applied to extract the region of interest (i.e. prostate area) in the T2W images [19]. U-Net has shown strong performance in prior studies involving segmentation in MRI data. Prostate segmentation was necessary because without it the model also takes into account the unrelated organs and tissues surrounding the prostate. This causes noise and affects the performance of the model. In addition, it aligns with the PI-RADS guidelines, which excludes the assessment of the peripheral zone in T2W images. The segmentation step resulted in a mask of the prostate region. The output image mask was cropped and resized to 224x224, then stacked to three channels to ensure compatibility with

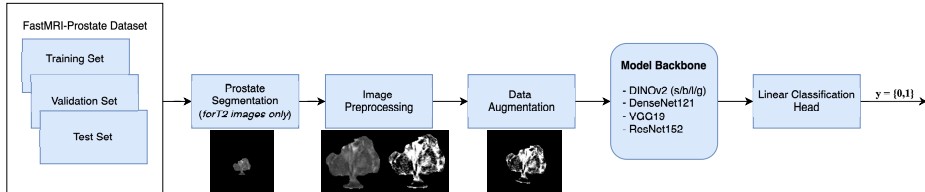

**Fig. 1.** End-to-end experiment pipeline. (a) Image preprocessing is composed of prostate segmentation (for T2W only), mask extraction, resizing, and normalization (b) Data augmentation involves translation, rotation, and horizontal flipping (c) The backbone weights are frozen (d) Linear classification head outputs 2 classes (0 = low risk, 1 = high risk)

DINOv2's expected input shape. Data augmentation methods such as horizontal flipping, random rotation between -10 to 10 degrees, and translation, with limited minimum and maximum values for each axis, to avoid accidentally removing the region of interest in the augmented samples. Afterwards, normalization was applied to the images. The same preprocessing and augmentation steps were performed for the DWI model, except for prostate segmentation. ADC maps and b1500 DWI images were stacked after applying the preprocessing methods.

### 2.3 Training

All models were trained using PyTorch 2.0 [18] on NVIDIA RTX A6000 GPUs. The DINOv2 models were obtained from the official DINOv2 website. There are four backbones that vary according to the parameters they have. The classification head was left by default to show that linear probing is sufficient for the downstream task of csPCa classification. To avoid overfitting, several methods were performed. Cosine learning rate annealing was applied to stabilize model training. We used SGD optimizer with an initial value for learning rate set at $1e^{-5}$. Due to the nature of the problem where most samples will generally be non-csPCa, a weighted binary cross entropy loss function was applied to take into account the class imbalance.

$$\mathcal{L} = -\left(w_1 \cdot y \cdot \log(\hat{y}) + w_0 \cdot (1-y) \cdot \log(1-\hat{y})\right)$$

Where $\mathcal{L}$ is the loss function:

- $w_1$ is the weight of the majority class
- $w_0$ is the weight of the minority class
- $y$ is the true label
- $\hat{y}$ is the predicted label

The DenseNet121, VGGNet19, and ResNet152 models were obtained from the collection of readily available PyTorch models. These were already pretrained on the ImageNet [6] dataset. The model backbones were used as a feature extractor while a linear classification head was trained in the same manner as

DINOv2 models to minimize the variation between the two groups. All of the models were trained with 20 epochs and a batch size of 32. The batch size was chosen as the maximum that can fit in the GPU resource used for this study. The only current model, as of writing, that has set an initial baseline performance for FastMRI Prostate dataset is the ConvNeXt architecture pretrained also on ImageNet. For each of these models, a linear classifier was trained on top of frozen pretrained features to evaluate downstream task performance. The end-to-end training pipeline is shown in Figure 1.

Implementation code will be made available upon acceptance to ensure reproducibility.

## 3   Results and Discussion

### 3.1   Evaluation

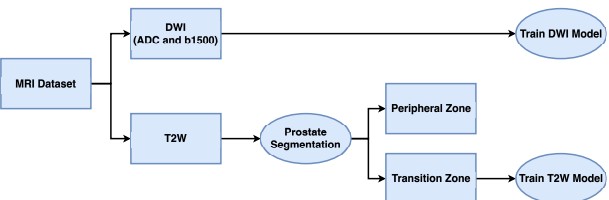

**Fig. 2.** Simplified flowchart on PI-RADS based on the MRI sequence

Given the biparametric MRI input from FastMRI Prostate, two models were trained based on the zone of the prostate. This strategy was patterned after the PI-RADS assessment standards which suggests the use of T2W in assessing the transition zone and DWI for the peripheral zone. Each model will be able to focus on the characteristics of each MRI sequence. Final score is then evaluated in the same manner as clinicians do when referencing PI-RADS. Figure 2 shows the PI-RADS v2 assessment guidelines that clinicians use in assessing a prostate MRI.

**Table 2.** Comparison of AUC performance across all models

|       | DINOv2 | | | | CNN | | | |
|-------|-------|-------|-------|-------|------------|--------------|--------|---------|
|       | ViT-S | ViT-B | ViT-L | ViT-g | ResNet-152 | DenseNet-201 | VGG-19 | ConvNeXt |
| **T2W** | 0.83 | 0.86 | 0.882 | **0.889** | 0.83 | 0.8 | 0.77 | 0.83 |
| **DWI** | 0.797 | 0.83 | 0.86 | **0.862** | 0.75 | 0.79 | 0.71 | 0.8 |

The model was evaluated using the area under the receiver operating characterstic curve (AUROC) in order to find out how well the model is able to

classify. The task is approached as a binary classification problem, PI-RADS labels greater than or equal to 3 show high risk for csPCa requiring biopsy or other follow-up, and PI-RADS less than 3 indicates low risk for csPCa. This aligns with the clinically relevant threshold for distinguishing non-suspicious from suspicious findings. The developed model was tested against the test set of the FastMRI Prostate dataset and compared its performance against ImageNet pretrained CNN models.

### 3.2    Results

Table 2 shows that overall performance of DINOv2 pretrained models obtain a higher AUROC than the CNN-based models. An explanation for this is the quality of features learned by DINOv2 models during pretraining on a huge amount of unlabeled data, through self-supervised learning, compared to ImageNet1k. Specifically, the ViT-g variant obtained the highest score among all other methods for both the T2W model and DWI model. In comparison, the ViT-S variant scored similarly with other CNN-based models. Despite its significantly lower number of parameters (20M), it still achieves performance on par with ResNet152 (60M). It is expected that as the number of parameters increases, the performance also improves but only until a certain point. This can be observed with the score of ViT-L and ViT-g.

**Table 3.** Training (in minutes) and Inference Time (in milliseconds) for all DINOv2 model sizes including their corresponding number of parameters (in millions) and giga floating-point operations per second (GFLOPs) per image.

| MRI Seq. | Variant | Params | GFLOPs | Training Time | Inference Time |
|---|---|---|---|---|---|
| T2W | ViT-S/14 | 20 | $\approx 4.5$ | 1.77 | 0.78 |
| | ViT-B/14 | 86 | $\approx 17$ | 3.01 | 1.33 |
| | ViT-L/14 | 300 | $\approx 61$ | 6.02 | 2.66 |
| | ViT-g/14 | 1,100 | $\approx 225$ | 15.23 | 6.72 |
| DWI | ViT-S/14 | 20 | $\approx 4.5$ | 4.68 | 0.62 |
| | ViT-B/14 | 86 | $\approx 17$ | 9.5 | 1.28 |
| | ViT-L/14 | 300 | $\approx 61$ | 32.5 | 4.44 |
| | ViT-g/14 | 1,100 | $\approx 225$ | 118.2 | 16.21 |

Table 3 shows the wall-clock training and inference time for all DINOv2 models under the same hardware and software environment to ensure fair and consistent measurement. The time measurement includes the end-to-end pipeline from loading the image up to running the validation and test set evaluation. There is minimal performance difference between the two models yet there is a two to four-fold difference in the training and inference time.

In Table 4, we show the effect of performing preprocessing on the model AUC scores. Ideally, DINOv2 features are ready to use for natural scene images. However, in the context of medical imaging, there is a domain shift that needs to be

**Table 4.** Performance of the ViT-g model after individual preprocessing steps applied independently to T2W and DWI modalities. The baseline indicates performance without any preprocessing. Scores reflect AUC performance, and the absolute improvement compared to the baseline for each MRI sequence.

| MRI Seq. | Preprocessing Step | AUC | Improvement |
|---|---|---|---|
| T2W | Baseline | 0.720 | – |
| | Center crop | 0.833 | 0.113 |
| | Zone segmentation + mask | 0.865 | 0.145 |
| | Horizontal flipping | 0.840 | 0.120 |
| | Rotation | 0.840 | 0.120 |
| | Translation | 0.830 | 0.110 |
| | Normalization | 0.872 | 0.152 |
| | **All** | **0.889** | **0.169** |
| DWI | Baseline | 0.821 | – |
| | Horizontal flipping | 0.825 | 0.004 |
| | Rotation | 0.826 | 0.005 |
| | Translation | 0.830 | 0.009 |
| | Normalization | 0.847 | 0.026 |
| | **All** | **0.862** | **0.041** |

addressed. We explore if DINOv2 as a feature extractor on a linear classification head is capable of differentiating between csPCa and non-csPCa. For the T2W model, the baseline score is 0.72. Applying all preprocessing steps resulted in 0.169 increase which can be attributed to applying prostate zone segmentation (U-Net) and normalization. The T2W images need to be segmented as there are several organs visible in this MRI sequence. For this downstream task, we only need the prostate area, and referencing how clinicians use PI-RADS, the primary focus of T2W is for the transition zone only. Thus, it only makes sense to extract only the transition zone area of the prostate. In the DWI model, there is minimal improvement in the baseline (0.82) with just an increase of 0.041 after applying the preprocessing steps. Performing normalization contributed the most to the improvement as it standardizes the intensity distribution resulting in a better signal-to-noise ratio. This aligns with the PI-RADS as it focuses on the hypointensity for ADC and hyperintensity of high b-value DWI images. Despite adding a few preprocessing steps, this does not significantly affect the time it takes to train the model or perform inference.

### 3.3   Limitations and Future Work

This study utilized minimal preprocessing and a simple classification head to evaluate the capabilities of feature extractors, particularly DINOv2's backbones, without extensive customization. The scope was limited to binary classification to establish the potential of DINOv2 for prostate cancer detection on bpMRI. Additionally, while this study focuses on linear probing, fine-tuning offers a promising avenue for future improvement. Future work will expand to multiclass classification using the full PI-RADS scoring system (1–5). While the T2W

model showed promising results, its performance may improve with advanced prostate segmentation techniques. Beyond the benchmark dataset, we are now evaluating the method on clinical MRI data from a tertiary hospital to assess the method's performance in real-world settings and its potential for clinical use.

## 4  Conclusion

This study highlights the advantages and limitations of DINOv2 in 2D binary classification when applied to FastMRI Prostate. The ViT-g variant obtained the highest AUROC for both T2W and DWI models. Despite freezing the backbone which allowed it to act as a feature extractor, it still achieved strong performance. However, training required significant VRAM ($\approx$ 46GB). Given minimal performance differences, the ViT-L variant is a more resource-efficient alternative, using roughly half the VRAM of ViT-g. A key strength of the DINOv2 models lies in the high-quality feature representations learned during large-scale pretraining. Future work will extend to 2D multi-class classification across all PI-RADS classes using the FastMRI Prostate dataset.

**Acknowledgments.** This study was funded by the University of the Philippines Intelligent Systems Center (ISRGA2024-24).

**Disclosure of Interests.** The authors have no competing interests to declare that are relevant to the content of this article.

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
