# OpenReview forum: "Self-supervised Vision Transformers  for Prostate Cancer Classification in Biparametric MRI"
_MICCAI.org/2025/Workshop/MSB_EMERGE — MSB EMERGE 2025 Conditionalrequiresmajorrevision_

### Official Review · Reviewer_R4o1 · 2025-07-08

**Recommendation:** 3
**Confidence:** 3

**Clarity:**

The paper has significant clarity issues that hinder understanding, substantial revision is required to improve clarity

**Feedback:**

- Multi-Class Classification: Extend experiments to predict all PI-RADS grades (1–5) to fully align with clinical reporting standards.

- Fine-Tuning Evaluation: Compare linear probing with partial/full backbone fine-tuning to assess potential performance gains.

- External Dataset Validation: Test on independent clinical bpMRI datasets to confirm model generalizability.

- Consistent Preprocessing: Apply prostate segmentation uniformly to both T2W and DWI to ensure modality-consistent preprocessing.

- Class Imbalance Strategies: Explore additional imbalance handling techniques (e.g., oversampling, undersampling, SMOTE) beyond weighted loss.

**Justification:**

This study makes a significant contribution by adapting a cutting-edge self-supervised ViT to prostate MRI classification, demonstrating strong performance and detailed efficiency analysis. However, its scope is limited by binary classification, frozen backbones, and single-dataset validation.

**Reproducibility:**

Sufficient amount of details available for reproducing the main results, and open access is provided (or promised upon acceptance) to source code and/or data

**Strengths:**

- State-of-the-Art Model Application: First application of DINOv2 ViT to medical imaging classification, showcasing superior feature representations over traditional CNNs.

- Benchmark Establishment: Provides a clear baseline on FastMRI Prostate with slice-level PI-RADS labels, facilitating future comparative studies.

- Performance vs. Efficiency Analysis: Reports both AUROC and computational metrics (training/inference time, VRAM) across ViT-S to ViT-g, guiding model selection under resource constraints.

- Preprocessing Impact Study: Details the contribution of segmentation and normalization steps to AUC improvements, underscoring the importance of domain-specific preprocessing.

**Summary:**

This paper evaluates the performance of DINOv2-based self-supervised Vision Transformer (ViT) models for clinically significant prostate cancer (csPCa) classification using the FastMRI Prostate public bpMRI dataset with PI-RADS labels. For both T2-weighted (T2W) and diffusion-weighted imaging (DWI; ADC map + b1500) sequences, the authors train and validate ViT variants (ViT-S, ViT-B, ViT-L, ViT-g) via linear probing, and compare against ImageNet-pretrained CNNs (ResNet-152, DenseNet-201, VGG-19, ConvNeXt). The largest model, ViT-g, achieves AUROC of 0.889 on T2W and 0.862 on DWI, demonstrating superior compute-performance trade-offs; notably, ViT-S (21M parameters) matches ResNet-152 (60M) performance. Training/inference time and VRAM usage are also quantitatively analyzed for each variant.

**Weaknesses:**

- Binary Classification Only: Limits to PI-RADS ≥3 vs. <3, without exploring full multi-class PI-RADS 1–5 classification, thus not fully reflecting clinical reporting standards.

- Backbone Frozen: Only linear probing is performed; backbone fine-tuning is not evaluated, leaving potential performance gains unexplored.

- Single Dataset Validation: Experiments conducted solely on FastMRI Prostate; external clinical datasets are not tested, so generalizability remains uncertain.

- Inconsistent DWI Preprocessing: Prostate segmentation applied to T2W but omitted for DWI, leading to inconsistent preprocessing across modalities.

---

### Official Review · Reviewer_h6QD · 2025-07-08

**Recommendation:** 4
**Confidence:** 4

**Clarity:**

The paper is clear and well-written, with minor areas for improvement in clarity

**Feedback:**

Add class imbalance information(based on scoring between 1-5) to Table 1 and ensure consistent number formatting(see 1399).

Clarify Table 2 by specifying the metric, explaining highlights, and adding variance/quantile intervals.

 Address/discuss the ordinal nature of PI-RADS rather than reducing to binary classification.

 Remove redundant Table 3; use color highlighting in Table 2 instead.

 Improve Table 4’s caption and include model parameters and FLOPs.

 Clarify Table 5’s notation and improve its caption to explain the evaluation procedure.

 Report additional metrics (e.g., balanced accuracy, AU-PR) in all evaluation tables.

 Reconsider and justify the choice of evaluation baselines.

**Justification:**

Despite strong motivation, solid results, and a reproducible approach, the paper has several presentation and methodological weaknesses—especially regarding evaluation metrics, table clarity, and the limited framing of the classification task. These issues limit clinical relevance and interpretability but are mostly addressable in revision. Given the overall status of the paper, a weak accept is justified.

**Reproducibility:**

Sufficient amount of details available for reproducing the main results, and open access is provided (or promised upon acceptance) to source code and/or data

**Strengths:**

Well-written and well-motivated introduction, supported by relevant related work.

Decent benchmark DINOv2 models on the FastMRI Prostate dataset.

Demonstrates DINOv2’s superior performance over CNNs for this task.

Promises code release for reproducibility.

Mostly thorough experimental comparison across multiple model sizes and architectures.

Highlights computational efficiency and resource trade-offs.

**Summary:**

This paper evaluates DINOv2 vision transformer models for classifying clinically significant prostate cancer using the FastMRI Prostate bpMRI dataset with PI-RADS labels. DINOv2 models outperform traditional CNNs, offering a strong balance between accuracy and computational efficiency, and establish new baselines for this public dataset.

**Weaknesses:**

Only binary classification is performed, ignoring the ordinal nature of PI-RADS, which limits clinical applicability.

Tables are a mess, lacking proper captions and are uninformative:

Table 1 does not report class imbalance, especially for ordinal ranking. Inconsistent formatting in Table 1 (missing comma in “1399”).

Table 2 lacks clarity on the metric used and the meaning of highlighting; does not report variance or quantile intervals.

Table 3 largely repeats Table 2; redundant rather than informative.

Table 4’s caption could be improved and should include parameters and FLOPs.

Table 5 uses unclear notation (“-”); unclear if augmentations are cumulative or independent; caption needs improvement.

Only AUC is reported, which is susceptible to class imbalance; no balanced accuracy or AU-PR provided.

Baseline choice for evaluation is questionable; potentially better baselines exist or more fair comparisons in terms of compute should be done.(i.e. wallclock)

---

### Official Review · Reviewer_jdtp · 2025-07-08

**Recommendation:** 2
**Confidence:** 3

**Clarity:**

The paper is generally clear but has some clarity issues that could be addressed with moderate revision

**Feedback:**

- The papers need to be improved for clarity. Some parts of the introduction are confusing and hard to read. Further, the lack of proper citing in the paper is also making the paper harder to read. Also, Table 3 can be deleted as it repeats the results from Table 2.
- In a paper that states the importance of including the PI-RAD score in the dataset, it would be necessary to perform multiclass classification rather than simplifying the problem to a binary classification. Including the results of both to see the change in the performance could also be an interesting addition to the paper.
- The contributions of the paper need to be improved. Currently, the paper is simply evaluating a new dataset on pre-existing methods.

**Justification:**

The paper is not written clearly and misses citations. The contributions of the paper are very limited.

**Reproducibility:**

Sufficient amount of details available for reproducing the main results, and open access is provided (or promised upon acceptance) to source code and/or data

**Strengths:**

- The methods are described in sufficient detail to allow replication.
- The methodology, while straightforward, is technically sound and free of major errors.
- The addressed problem of prostate cancer risk classification is an important clinical issue.

**Summary:**

The paper is evaluating a number of existing pre-trained models on the task of binary classification of low vs. high risk of prostate cancer in T2-weighted and diffusion-weighted MR.

**Weaknesses:**

- The paper is missing citations, e.g., not citing the FastMRI Prostate dataset in the introduction section, never citing ImageNet. In Sec. 2.3 authors mention that ConvNext architecture already set an initial baseline for the FastMRI Prostate dataset, but no citation is included.
- The paper is not written clearly with missing citations.
- No real contribution of the work. We already know that features from DINOv2 are transferable for medical tasks, which the authors mentioned in the paper.
- The inclusion PI-RAD score is discussed as one of the main novelties of the FastMRI Prostate dataset. It is described that the dataset includes ratings from 1-5, but the evaluation is only completed on the binary classification task, stating multiclass classification is left for future work.
- Fig. 2 the text is too small, makes the figure hard to read.
- Table 3 is unnecessary, it repeats the results from Table 2.
- The evaluation is only done on linear probing. It would be interesting to compare the results of linear probing to fine-tuning.

---

### Decision · Program_Chairs · 2025-07-18

**Decision:**

Conditional Accept (requires major revision)

**Comment:**

The paper is conditionally accepted to the EMERGE workshop. While the reviewers recognize the value of the contribution, we ask the authors to clearly state the limitations and adjust the presentation of novelty accordingly in the camera-ready version.

Acceptance is contingent on satisfactorily addressing these concerns.